# On the Relationship between
# And-Sum Circuits and Deterministic Boolean Circuits

**Jonas R. L. Gonçalves**[1]                    **Denis D. Mauá**[1]

[1]Computer Science Dept., Institute of Mathematics and Statistics, University of São Paulo, São Paulo, São Paulo, Brazil

## Abstract

And-Sum Circuits are a recently proposed target representation language for knowledge compilation that utilizes AND and SUM nodes with signed edges. A key property of these circuits is "booleanity", which states that every subcircuit computes a 0/1-valued function. We provide a characterization of booleanity in terms of logical restrictions on the inputs to SUM nodes, analogous to how determinism is characterized in traditional Boolean Circuits. We demonstrate that And-Sum Circuits where SUM nodes are connected by either at most one positive edge or by only positive edges can be efficiently translated to and from a semantically equivalent decomposable and deterministic Boolean Circuits. This implies that And-Sum Circuits whose SUM nodes have input degree at most two are expressively equivalent to decomposable and deterministic Boolean Circuits.

## 1 INTRODUCTION

Knowledge Compilation is the process of translating a knowledge base from a source language to a target language that allows for answering queries of a specific type exactly and tractably [Akers, 1978, Darwiche and Marquis, 2002]. When the type of query involves counting models, the target language typically considers Boolean Circuits that satisfy *determinism*, which roughly states that the inputs of any or-gate specify mutually exclusive properties [Amarilli and Capelli, 2024]. Essentially, determinism ensures that the model count of a disjunction of formulae be computed as the sum of the model count of each subformula: $\mathrm{MC}(\alpha \vee \beta) = \mathrm{MC}(\alpha) + \mathrm{MC}(\beta)$ if $\alpha \wedge \beta = \bot$. Enforcing determinism however may cause a blow up of the minimum size of the target representations.

Recently, Onaka et al. [2025] proposed *Boolean Decom-posable And-Sum Circuits* (B-DASC), a target representation language that employs AND ($\wedge$) and SUM ($+$) nodes with signed edges. The inclusion of negative edges and addition allows for representing model subtraction, for instance, when one models the model count of a disjunction as $\mathrm{MC}(\alpha \vee \beta) = \mathrm{MC}(\alpha) + \mathrm{MC}(\beta) - \mathrm{MC}(\alpha \wedge \beta)$. As a consequence, B-DASCs allow for efficient model counting without requiring *determinism*. They are also closed under negation, and allow for efficient conjoin operation [Darwiche, 2011]. Hence, B-DASC are a very interesting class of target representations to study.

*Booleanity* in B-DASCs refers to the property that any subcircuit computes a Boolean function, that is, that the output of any gate is 0/1-valued for any assignment of its inputs. While such property is trivially satisfied by Boolean Circuits, the use of addition and negative weights in And-Sum Circuits makes enforcing such property less trivial.

In this work we study in greater detail the effects of requiring booleanity for And-Sum circuits, and its relation with deterministic Boolean Circuits. In particular, we characterize booleanity in terms of logical restrictions of the underlying Boolean formulae, similarly to how determinism can be characterized. We show that any B-DASC whose gates are connected by at most one negative or by only positive edges can be translated to semantically equivalent deterministic and decomposable Boolean circuits. And we show the converse to also be true. Those results provide a more clear view of B-DASC in the knowledge compilation landscape.

## 2 CIRCUIT REPRESENTATIONS

A *Boolean Circuit* is a directed acyclic graph whose inner nodes are labeled as $\ominus$ (negation), $\wedge$ (conjunction) or $\vee$ (disjunction) gates, and whose leaves are associated with propositional variables (typically denoted as $X$, $Y$, etc.). We only consider here rooted circuits, that is, circuits with a single output node. A Boolean circuit thus represents a Boolean formula, obtained by interpreting each node as its

*Accepted for the 8th Workshop on Tractable Probabilistic Modeling at UAI (TPM 2025).*

logical function when the variables in the leaves are fixed at some input assignment.

*And-Sum Circuits* deviate from such traditional circuits by assigning inner nodes to either ∧ or ⊕, and by labeling edges connecting ⊕-nodes to their inputs with weights in $\{-1, 1\}$ [Onaka et al., 2025]. Given an assignment $\sigma$ to its input variables (leaves), an And-Sum Circuit computes a formula $f(\sigma)$ such that $f_v(\sigma) = \sum_c w_{vc} f_c(\sigma)$, if $v$ is a ⊕-node with inputs $\{f_c\}$ and weights $w_{vc}$, and $f_v(\sigma) = \min_c f_c(\sigma)$, if $v$ is a ∧-node with inputs $f_c$.

We say that a node in a Boolean or a And-Sum Circuit is *binary/ternary* if it has has at most two/three inputs. The *scope* of a (sub)circuit is the set of variables appearing at its leaves. We denote the scope of a circuit rooted at a node $v$ by $\mathrm{Var}(v)$.

The tractability of query answering with Circuits depends on a few properties, which we review next.

Negated Normal Form (NNF) Boolean circuits constrain ⊖-nodes to have only variable nodes (leaves) as input [Darwiche, 2001]; in effect, NNFs are generally represented having only ∧ and ∨ inner nodes and literals as leaves (e.g., $X$ or $\neg X$).

A Boolean (resp., And-Sum) circuit is *smooth* if for every ∨-node (resp. ⊕-node) it follows that any two input circuits have identical scopes, that is, $\mathrm{Var}(c) = \mathrm{Var}(c')$ for any two inputs $c$ and $c'$. A Boolean or And-Sum Circuit is *decomposable* if the scopes of any two inputs $c$ and $c'$ of a ∧-node are disjoint: $\mathrm{Var}(c) \cap \mathrm{Var}(c') = \emptyset$. Decomposability allows for tractable model satisfiability in Boolean circuits.

The *support* of a (Boolean or And-Sum) circuit is the set of assignments for which the respective function evaluates to non-zero. A Boolean (resp., And-Sum) Circuit is *deterministic* if the supports of the inputs of a ∨-node (resp., ⊕-node) are disjunct: $f_c(\sigma) \neq 0 \Rightarrow f_{c'}(\sigma) = 0$ for all input $c' \neq c$. This implies that $\sum_c f_c(\sigma) \leq 1$ for Boolean Circuits and that $\sum_c f_c(\sigma) \in \{-1, 0, 1\}$ for And-Sum Circuits. Determinism allows for tractable model counting in Boolean circuits.

A *vtree* is a rooted binary tree whose leaves correspond to variables (generally, of a circuit) [Pipatsrisawat and Darwiche, 2008]. Each node of a vtree thus corresponds to a binary partition of a subset of variables. A Boolean or And-Sum Circuit with binary ∧-nodes is *structured decomposable* if any ∧-node can be mapped to a node of the vtree such that the scopes of each input corresponds to the binary partition induced by the vtree node. Structured Decomposability allows for tractable *conjoin* operations, which takes two circuits $\alpha$ and $\beta$ and in polynomial time and space produces a circuit that represents the function $\alpha \wedge \beta$.

Decomposable Boolean And-Sum Circuits allow for tractable conjoin and model counting. Remarkably, the adop-

tion of signed edges also allows for tractable negation, a feature that is lacking in popular languages like smooth, deterministic and (structured) decomposable NNFs [Vinall-Smeeth, 2024]. For instance, the negation of an And-Sum Circuit $\phi$ can be represented by a circuit with a binary ⊕ root node with a child that always evaluate to 1, connected with a positive weight, and a child that is $\phi$, connected with a negative weight. That is, by computing $1 - \phi$. Moreover, conjoin and negation preserve booleanity, smoothness, and (structured) decomposability. Also, by composing negation and conjoins we also have available a disjoin operation for Boolean And-Sum Circuits [Onaka et al., 2025].

# 3 CHARACTERIZING BOOLEANITY

And-Sum Circuits are *Boolean* if every node outputs a value 0 or 1 for every input assignment, which suffices for ensuring that the circuit actually computes a Boolean function.

While Onaka et al. [2025] considered only Boolean decomposable And-Sum Circuits (B-DASCs), they did not care to characterize the necessary and sufficient conditions for this property to hold. We now investigate such necessary and sufficient conditions for booleanity in detail.

To simplify notation, for any ⊕-node $v$, we write $\psi_i^+$, $i = 1, \ldots, n$, to denote the input nodes connected by positive edges, and $\psi_j^-$, $j = 1, \ldots, m$, to denote the input nodes connected by negative edges. Thus, the function computed by the node $v$ is:

$$f_v(\sigma) = \sum_{i=1}^{n} \psi_i^+(\sigma) - \sum_{j=1}^{m} \psi_j^-(\sigma). \tag{1}$$

Since we assume the circuit is Boolean, we can also view each circuit as a Boolean function, used to form logical expressions (e.g., $\psi_1^+ \Rightarrow \psi_3^-$).

## 3.1 ALL EDGE WEIGHTS ARE POSITIVE

Suppose $m = 0$. Then, the function computed by a circuit root at $r$ is $f_r = \sum_{i=1}^{n} \psi_i^+$. For $f_r$ to be Boolean, $f_r(\sigma)$ must be either 0 or 1. If $f_r(x) = 0$, then $\psi_i^+(\sigma) = 0$ for all $i$. On the other hand, if $f_r(\sigma) = 1$, then exactly one $\psi_i^+(\sigma)$ must be 1, and all other children must be 0. This means that for any assignment $\sigma$, the children are *logically contradictory* (or mutually exclusive). That is, a ⊕-node with only positive weights is Boolean only if it is deterministic.

## 3.2 EXACTLY ONE POSITIVE EDGE WEIGHT

Now, suppose that there is exactly one child with a positive edge weight (i.e., $n = 1$ and $m > 0$), and any other children have negative edge weights. Then the function computed by $r$ is $f_r(\sigma) = \psi_1^+(\sigma) - \sum_{j=1}^{m} \psi_j^-(\sigma)$. It is immediately

clear that $f_r(\sigma) \leq 1$. Thus, we must only establish sufficient and necessary conditions for $f_r(\sigma) \geq 0$ in order for it to be Boolean. Those conditions are exactly given by $\psi_j^- \Rightarrow \psi_1^+$ for $j = 1, \ldots, m$, and by $\psi_j^- \Rightarrow \neg\psi_k^-$ for any $j \neq k \in \{1, \ldots, m\}$. Interestingly, those conditions constrain the boolean function represented by $f_r$ to be logically equivalent to

$$\psi_1^+ \wedge \bigwedge_{j=1}^m \neg\psi_j^- \Leftrightarrow \neg\left(\neg\psi_1^+ \vee \bigvee_{j=1}^m \psi_j^-\right). \quad (2)$$

Note that the clause on the right is deterministic by the reasoning above. We will exploit that fact to derive a translation to deterministic Boolean circuits later.

## 3.3 EXACTLY ONE NEGATIVE EDGE WEIGHT

Suppose now that $n > 1$ and $m = 1$. Thus the circuit computes $f_r(\sigma) = \sum_{i=1}^n \psi_i^+(\sigma) - \psi_1^-(\sigma)$. Clearly, in order to observe booleanity, we must have that $\psi_i^+(\sigma) + \psi_j^+(\sigma) \geq \psi_1^-(\sigma) \geq 0$ for any $i \neq j$ and $\sigma$. That is, we have that no more than two positively signed inputs can be true for the same configuration, that if exactly two positively signed inputs are true then the negatively signed input must also be true, and that if the negatively signed input is true then some positively signed input must be true as well. Those conditions imply that the Boolean function represented by $f_r$ is logically equivalent to

$$\bigvee_{i=1}^n \phi_i \vee \bigvee_{i=1}^n \bigvee_{j \neq i} \phi_{ij} \quad (3)$$

where

$$\phi_i = \psi_i^+ \wedge \bigwedge_{j \neq i} \neg\psi_j^+ \wedge \neg\psi_1^-, \quad (4)$$

and

$$\phi_{ij} = \psi_i^+ \wedge \psi_j^+ \wedge \bigwedge_{k \neq i, k \neq j} \neg\psi_k^+ \wedge \psi_1^-. \quad (5)$$

Unlike the case of exactly one positive edge weight, the formulas for $\neg\phi_i$ and $\neg\phi_{ij}$ are not deterministic clauses.

Consider the case of a ternary $\oplus$-node $r$, that is, when $n = 2$ and $m = 1$. Then the logical constraints above imply that $f_r$ is logically equivalent to

$$(\neg\psi_1^+ \wedge \psi_2^+ \wedge \neg\psi_1^-) \vee (\psi_1^+ \wedge \neg\psi_2^+ \wedge \neg\psi_1^-) \vee (\psi_1^+ \wedge \psi_2^+ \wedge \psi_1^-).$$

If $\psi_1^- \Leftrightarrow \psi_1^+ \wedge \psi_2^+$, then $f_r$ computes $\psi_1^+ \vee \psi_2^+$, even when $\psi_1^+$ and $\psi_2^+$ are not deterministic. If instead $\psi_1^- \Leftrightarrow \psi_1^+ \vee \psi_2^+$, then $f_r$ computes $\psi_1^+ \wedge \psi_2^+$, even when $\psi_1^+$ and $\psi_2^+$ are not decomposable.

## 3.4 POSITIVE AND NEGATIVE EDGE WEIGHTS

Suppose that $n > 0$ and $m > 1$. Then for Equation (1) to define a Boolean function, we need to ensure that whenever a subset of $p$ of the positively signed inputs are 1, then at least $p - 1$ negatively signed inputs must also be 1. That is, for $p = 2, \ldots, m$:

$$\bigvee_{S \subset [n]: |S| = p} \bigwedge_{i \in S} \psi_i^+ \implies \bigvee_{S \subseteq [m]: |S| = p-1} \bigwedge_{j \in S} \psi_j^-, \quad (6)$$

where $[k] = \{1, \ldots, k\}$. Conversely, if $p$ of the negatively signed inputs are 1, then at least $p$ of the positively signed inputs must be 1 as well. Thus, for $p = 1, \ldots, m$:

$$\bigvee_{S \subset [m]: |S| = p} \bigwedge_{j \in S} \psi_j^- \implies \bigvee_{S \subset [n]: |S| = p} \bigwedge_{i \in S} \psi_i^+. \quad (7)$$

The above conditions are necessary and sufficient for booleanity in such circuits.

As with the previous case, we do not obtain a formula represented as a (negation of) deterministic disjunction here.

$\oplus$-nodes with two or more positive edges and two or more negative edges can represent functions which are not expressed with a single positive edge or a single negative edge. For example, if $\psi_1^- \Leftrightarrow \psi_2^- \Leftrightarrow \psi_1^+ \wedge \psi_2^+$, then a $\oplus$-node with children $\psi_1^+, \psi_2^+, \psi_1^-$ and $\psi_2^-$ computes an XOR of $\psi_1^+$ and $\psi_2^+$.

# 4 RELATION TO DETERMINISTIC DECOMPOSABLE BOOLEAN CIRCUITS

The main difference of deterministic Decomposable Boolean Circuits (det-Ds) with respect to Negation Normal Form circuits is to allow negation also outside of literals, while preserving correctness of weighted model counting, for any ring [Monet and Olteanu, 2019]. Thus, det-Ds are naturally closed under negation. As Onaka et al. [2025] have shown, the same is true for Boolean And-Sum Circuits, as they allow the encoding of negation $\neg\phi$ as $1 - \phi$. This simple observation suffices to show that any det-D Circuit can be tractably transformed to a Boolean Decomposable And-Sum Circuit simply by transforming each $\ominus$-node into a $\oplus$-node with children $+1$ and $-\phi$, then transforming each $\vee$-node to a $\oplus$-node with the same children and positive edges. Due to determinism, $\vee$-nodes can be translated directly to $\oplus$-nodes with positive edges. The converse is also true. Because $\oplus$-nodes with only positively-signed edges are deterministic, they can be easily transformed into deterministic $\vee$ nodes. Accordingly, (smooth) And-Sum Circuits with only positively-signed edges can be tractably translated to (smooth) det-Ds, simply by mapping $\oplus$ into $\vee$ nodes.

A (smooth) Boolean And-Sum Circuit with exactly one positively-signed child at each $\oplus$-node can also be translated into a (smooth) det-D Circuit by applying the transformation in Figure 1 to every $\oplus$-node. The transformation is based on the fact that right hand formula in Expression (2) is a deterministic disjunction.

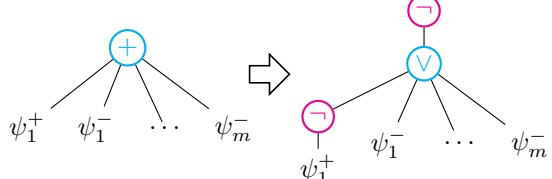

Figure 1: Transformation of a Boolean And-Sum Circuit with at most one positive subcircuit at each ⊕-node to a deterministic and decomposable Boolean Circuit.

Note that a binary ⊕-node (i.e., with two children) either contains only positive edge weights or exactly one positive edge weight. Thus, And-Sum Circuits with binary ⊕-nodes can be tractably translated into det-D Circuits.

Instead, ⊕-nodes with three children allow for nodes with two positive edge weights and one negative weights, that, as we discussed, can already encode non-deterministic disjunctions of their input. And ⊕-nodes with higher in-degree can express even more Boolean functions of their input. It remains an open question whether such nodes still allow for tractable translations to det-Ds, or whether they make Boolean And-Sum Circuits able to encode Boolean functions exponentially more succinctly than det-Ds.

We have thus shown that (smooth) decomposable And-Sum Circuits whose sum nodes have either only positively-signed children or at most one positively-signed child can be tractably transformed into (smooth) decomposable and deterministic Boolean Circuits. We call such a class *quasi-deterministic* And-Sum Circuits, as we have shown they typically define ⊕-nodes with are deterministic or near deterministic. Boolean And-Sum Circuits with binary ⊕-nodes are thus a subclass of such a language.

The diagram in Figure 2 updates the current state of knowledge about circuit equivalences with these new results. Arrows indicate the existence of a tractable (i.e., linear) transformation from the source language to the target language. Colored arrows are contributions of this work. The language q-b-DASC in the diagram refers to the class of quasi-deterministic And-Sum Circuits.

An interesting consequence of Figure 2 concerns the knowledge compilation framework based on structured decomposable And-Sum Circuits from [Onaka et al., 2025]. The authors detail a bottom-up compiler where ⊕-nodes arise solely from NOT operations, and their CONJOIN algorithm does not increase the arity of these nodes. This inherently limits all ⊕-nodes in their circuits to being binary. Since, as we have established (and Figure 2 confirms), And-Sum Circuits with only binary ⊕-nodes can be converted to structured det-Ds with linear overhead, it follows that the circuit class in [Onaka et al., 2025] is less general than the And-Sum formalism might suggest. By exclusively generating binary ⊕-nodes, their approach effectively confines itself

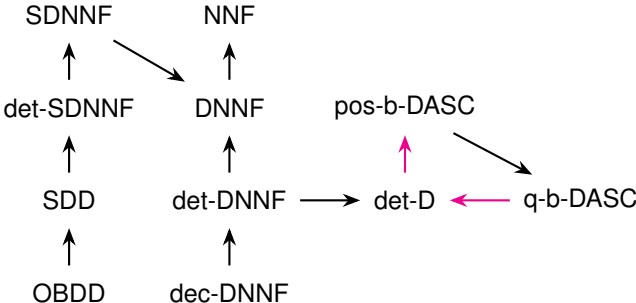

Figure 2: A Knowledge Compilation Map. s: smooth. d: deterministic. S: structured. D: decomposable. dec: Decision. pos: positive edges only. q: at most one positive edge or no negative edge at each sum node. And-Sum Circuits (ASC) are always considered Boolean. Arrows denote the existence of a linear-time transformations from the source to target language. Colored arrows are new results.

to the expressive power of structured det-Ds, thereby not fully exploring the representational capabilities that Boolean And-Sum circuits with higher-degree ⊕-nodes could offer.

## 5 CONCLUSION

And-Sum circuits are a recently proposed class of arithmetic circuits that uses arithmetic functions to compute Boolean functions. Like decomposable and deterministic circuits, they are closed under negation (even though they do not allow negation), and when satisfying structured decomposability, allow for efficient conjoin operation.

In this work, we have shown that the requirement that each node of an And-Sum circuit computes a Boolean function make such circuits behave similarly to deterministic Boolean circuits. In particular, we have characterized the necessary and sufficient logical constraints imposed by booleanity for different conditions: when sum nodes are connected via only positive edges, when they are connected via at most one positive edge (and the rest being negative), when they are connected via at most one negative edge (and the rest being positive), as well as the general case. We showed that the first two cases lead to conditions that can be easily represented in the language of decomposable deterministic Boolean circuits.

Thus, this work extends the current state of the knowledge compilation map, drawing new connections between arithmetic and Boolean circuits. The work leaves open the question of whether arbitrary boolean And-Sum Circuits can be tractably translated into deterministic Boolean circuits. As such, it also leaves open the characterization of the succinctness of (Boolean) And-Sum Circuits.

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
