# OpenReview forum: "On the Relationship between And-Sum Circuits and Deterministic Boolean Circuits"
_auai.org/UAI/2025/Workshop/TPM — TPM 2025_

### Official Review · Reviewer_5ni7 · 2025-06-15

**Rating:** 2

**Review:**

This paper considers And-Sum Circuits, which is a recently proposed knowledge compilation language.  Such circuits have decomposable and-nodes, and smooth and deterministic sum-nodes.  Notably. sum-nodes have children whose edges are annotated with +1 or -1.  The model count of a sum-node is correspondingly the weighted sum of its children.

The current paper more closely examines the relationship of And-Sum Circuits (with negative edge weights) and pure Boolean circuits.  They identify some conditions where and And-Sum Circuit can be converted to a deterministic decomposable circuit, and vice-versa.

There has been some recent interest in tractable probabilistic models with negative weights and parameters.  One example is the recent paper:

  Sum of Squares Circuits
  by Lorenzo Loconte, Stefan Mengel, Antonio Vergari
  in AAAI'2025

There are some other examples of TPMs and PGMs with negative weights and parameters.  There are also some results from the theory literature that show the impact that negations can have on the succinctness of circuits, e.g., Valiant's paper "Negation can be exponentially powerful."

At some level, the paper is examining conditions on when the power of negation is limited, and does not necessarily buy you anything.  I believe the topic of the paper is a very good fit for the TPM workshop.

---

### Official Review · Reviewer_mPsN · 2025-06-16
**Accept**

**Rating:** 3

**Review:**

This work, while simple, makes a concrete contribution to the the knowledge compilation map based on a recently proposed representation language. It is appropriate for the workshop and will be interesting and relevant to the community.

Minor typos:
- “those conditions constraint the boolean function” … “constrain” not “constraint”
- “or at most one positively-signed children” … should end with “child”